# miR-7 Regulates GLP-1-Mediated Insulin Release by Targeting β-Arrestin 1

**DOI:** 10.3390/cells9071621

**Published:** 2020-07-06

**Authors:** Alessandro Matarese, Jessica Gambardella, Angela Lombardi, Xujun Wang, Gaetano Santulli

**Affiliations:** 1Department of Medicine, Fleischer Institute for Diabetes and Metabolism (FIDAM), Einstein-Mount Sinai Diabetes Research Center (ES-DRC), Albert Einstein College of Medicine, New York, NY 10461, USA; alessandromatarese@yahoo.it (A.M.); jessica.gambardella@einsteinmed.org (J.G.); angela.lombardi@einsteinmed.org (A.L.); xujun.wang@einsteinmed.org (X.W.); 2AORN “Antonio Cardarelli”, 80100 Naples, Italy; 3Department of Advanced Biomedical Science, “Federico II” University, and International Translational Research and Medical Education Consortium (ITME), 80131 Naples, Italy; 4Department of Microbiology and Immunology, Albert Einstein College of Medicine, New York, NY 10461, USA; 5Department of Molecular Pharmacology, Albert Einstein College of Medicine, New York, NY 10461, USA

**Keywords:** β-arrestin 1, cAMP, diabetes, epigenetics, glucose-stimulated insulin secretion (GSIS), miRNA-7

## Abstract

Glucagon-like peptide-1 (GLP-1) has been shown to potentiate glucose-stimulated insulin secretion binding GLP-1 receptor on pancreatic β cells. β-arrestin 1 (βARR1) is known to regulate the desensitization of GLP-1 receptor. Mounting evidence indicates that microRNAs (miRNAs, miRs) are fundamental in the regulation of β cell function and insulin release. However, the regulation of GLP-1/βARR1 pathways by miRs has never been explored. Our hypothesis is that specific miRs can modulate the GLP-1/βARR1 axis in β cells. To test this hypothesis, we applied a bioinformatic approach to detect miRs that could target βARR1; we identified hsa-miR-7-5p (miR-7) and we validated the specific interaction of this miR with βARR1. Then, we verified that GLP-1 was indeed able to regulate the transcription of miR-7 and βARR1, and that miR-7 significantly regulated GLP-1-induced insulin release and cyclic AMP (cAMP) production in β cells. Taken together, our findings indicate, for the first time, that miR-7 plays a functional role in the regulation of GLP-1-mediated insulin release by targeting βARR1. These results have a decisive clinical impact given the importance of drugs modulating GLP-1 signaling in the treatment of patients with type 2 diabetes mellitus.

## 1. Introduction

The incretin glucagon-like peptide-1 (GLP-1) is a polypeptide hormone produced mainly in entero-endocrine L cells of the gut known to potentiate glucose-stimulated insulin secretion (GSIS) in pancreatic β cells [1,2]. GLP-1 action is mediated by GLP-1 receptors, a member of the seven-transmembrane family of G protein-coupled receptors (GPCRs) [3,4,5,6].

MicroRNAs (miRNAs, miRs) are small non-coding single-stranded ribonucleic acids (RNAs), highly conserved from plants to mammals, which are able to enhance messenger RNA (mRNA) degradation and/or inhibit protein translation by binding to the 3′-untranslated regions (UTRs) of target mRNAs [7,8,9,10,11,12,13]. They play crucial regulatory roles in a variety of biological processes, including regulation of differentiation, development, and function of β cells [14,15,16]. Specifically, recent studies have identified a number of miRs involved in the regulation of insulin release [17,18,19]. In terms of therapeutic potential, miRs represent a novel and very appealing strategy to manipulate metabolic processes as their activity can be efficiently modulated with RNA-based technologies [20,21,22].

Experimental evidence has recently shown that β-arrestin 1 (βARR1), a protein known to be involved in the regulation of signal transduction of GPCRs [23,24,25,26,27,28,29], plays a key role in the desensitization of GLP-1 receptor in β cells [30]. Therefore, targeting βARR1 by miR intervention could be a promising strategy for the treatment of diabetes mellitus. The aim of the present study was to identify a miR that targets βARR1 to modulate β cell function. A bioinformatic screen resulted in the identification of hsa-miR-7-5p (briefly noted as miR-7 in the rest of the paper) as potentially capable of repressing βARR1 mRNA expression. In our biological validation of this screen, miR-7 was found to actually target βARR1 and to be upregulated in β cells after GLP-1 stimulation. The mechanistic role of miR-7 was further confirmed by assessing the regulation of insulin release in β cells.

## 2. Materials and Methods

### 2.1. Cell Culture and Reagents

INS-1 β cells were cultured in a humidified atmosphere (37 °C) containing 5% CO_2_ in RPMI-1640 medium, and insulin secretion was assessed as we previously described and validated [31,32,33,34]. All experiments were performed using INS-1 β cells between the 20th and 40th passage. All reagents were from Millipore-Sigma (Burlington, MA, USA), unless otherwise stated.

### 2.2. Insulin Secretion

Insulin release in response to different stimuli was assessed using a commercially available enzyme-linked immunosorbent assay, following the manufacturer’s instructions (Mercodia, Uppsala, Sweden).

### 2.3. Cyclic AMP (cAMP) Assay

Intracellular cAMP content was measured by using the cAMP enzyme immunoassay kit (Enzo Life Sciences, Farmingdale, NY, USA), according to the manufacturer’s instructions; final cAMP concentrations per well were normalized by total protein as described.

### 2.4. Identification of miR-7 as a Regulator of βARR1

To identify miRs targeting the 3′-UTR of βARR1, we used online target prediction tools, including miRWalk-3 and Targetscan version 7.2, as we previously described [35,36]. These programs predict biological targets of miRs by searching for the presence of conserved 8mer and 7mer sites that match the seed region of miRs.

### 2.5. Biological Validation of miR-7 as a Regulator of βARR1

To assess the actual effects of miR-7 on βARR1 gene transcription, we used a luciferase reporter containing the 3’-UTR segment of the predicted miR interaction sites, both wild-type and mutated, in INS-1 cells. The mutant construct of *βARR1* 3′-UTR, carrying a substitution of two nucleotides within the predicted miR-*7* binding sites of *βARR1* 3′-UTR (see Figure 1A) was obtained using a commercially available site-directed mutagenesis kit (New England Biolabs, Ipswich, MA, USA), as we described [35]. Using Lipofectamine RNAiMAX (ThermoFisher Scientific, Waltham, MA, USA), cells were transfected with the 3′-UTR reporter plasmid (0.05 μg) and miR-7 mirVana^TM^ (50 nM) mimics or inhibitors (ThermoFisher Scientific) as well as a non-targeting negative control (scramble), according to the manufacturer’s instructions, as described [35]. Forty-eight hours after transfection, Firefly and Renilla luciferase activities were assessed using a commercially available Luciferase Reporter Assay System (Promega, Madison, WI, USA). Firefly luciferase was normalized to Renilla luciferase activity. Levels of miR-7 were measured using individual TaqMan miRNA assays, according to the manufacturer’s instructions; miR expression was normalized to the level of U6; standard TaqMan gene expression assays from Applied Biosystem were used, as we previously described and validated [35,36,37]. Cellular expression of βARR1 was determined by RT-qPCR, as we previously described [32,33,37], normalizing to endogenous glyceraldehyde 3-phosphate dehydrogenase (GAPDH). Sequences of oligonucleotide primers (Merck KGaA, Darmstadt, Germany) are presented in Appendix A.

### 2.6. Immunoblotting

Immunoblotting assays were performed, as previously described and validated by our group described [36,37,38,39,40] and developed with the Odyssey system (LI-COR Biosciences, Lincoln, NE, USA). The intensity of the bands was quantified by using the FIJI (Fiji Is Just ImageJ) software. Antibodies were purchased from Cell Signaling Technology (Danvers, MA, USA): CREB (catalog number: #4820), Phospho-CREB (pCREB Ser^133^; catalog number: #9198), ERK1/2 (catalog number: #9102); and from Santa Cruz Biotechnology (Dallas, TX, USA): p-ERK Antibody (catalog number: #sc-7383).

### 2.7. Statistical Analysis

Data are expressed as means ± standard error of means (SEM). Statistical analyses were performed in Prism (GraphPad Software, Version 8.0; Prism, San Diego, CA, USA). Statistical significance was tested using the nonparametric Mann–Whitney U test or two-way ANOVA followed by Tukey–Kramer multiple comparison test, as appropriate. Significant differences were established at a *p*-value < 0.05.

## 3. Results

### 3.1. βARR1 Is a Molecular Target of miR-7

Through bioinformatic analyses, we identified miR-7 as a potential regulator of βARR1. We selected miR-7 because it had been previously described as a key player in beta cell physiology. Specifically, miR-7 is considered to be a prototypical neuroendocrine miR, being highly expressed in neuroendocrine organs, including the endocrine pancreas and the pituitary and adrenal glands [41,42,43,44]. Moreover, the complementary nucleotides between the target region of βARR1 3’ untranslated region (3′-UTR) and miR-7 are evolutionarily highly conserved across different species, including humans, non-human primates, and rodents (Figure 1A). The proposed relationship was substantiated by an actual validation of seed complementarity, confirming the interaction between miR-7 and βARR1 3′-UTR in INS-1 β cells through a luciferase assay (Figure 1B).

### 3.2. GLP-1 Triggers miR-7 Transcription

Then, we tested the effects of GLP-1 on the transcription of both miR-7 and βARR1 in β cells. We found that 100 nM GLP-1 (a dose previously verified to be effective to induce insulin release in INS1 cells [30]) induced a significant upregulation of miR-7 and downregulation of βARR1 (Figure 2), 2 h post stimulation.

### 3.3. miR-7 Regulates GLP-1-Induced Insulin Secretion in β Cells

To further substantiate the functional aspects of our results, we evaluated the effects of miR-7 on GLP-1-mediated insulin release. We demonstrated that miR-7 significantly reduced GLP-1-induced GSIS in β cells (Figure 3A), whereas no significant effects were noted on KCl-induced insulin secretion (Figure 3B) or insulin content (Figure 3C). 

Strikingly, these alterations were rescued when β cells were treated with a specific inhibitor of miR-7 (Figure 3). 

We also verified that miR-7 mimic significantly decreased the transcription levels of βARR1 in β cells (Appendix A).

### 3.4. miR-7 Regulates GLP-1-Mediated cAMP Production in β Cells

Since GLP-1 has been shown to induce insulin release via an increased production of cyclic AMP (cAMP) [30,45], we verified the effects of miR-7 on the generation of cAMP in β cells. We found that miR-7 significantly decreased GLP-1-induced cAMP levels (Figure 4), whereas incubation with a specific miR-7 inhibitor markedly increased cAMP production (Figure 4).

### 3.5. miR-7 Modulates the GLP-1-Mediated Activation of ERK and CREB

Finally, since βARR1 knockdown has been shown [30] to reduce the phosphorylation of ERK and CREB, two factors downstream of GLP-1 signaling, we tested the effects of miR-7 on both these signaling pathways. We found that the GLP-1-induced activation of ERK and CREB was significantly attenuated by miR-7 (Appendix A).

## 4. Discussion

In the present study, we demonstrate, for the first time, that miR-7 plays a pivotal role in the regulation of GLP-1-mediated insulin release in β cells via a mechanism that involves its direct targeting of βARR1.

Our results are consistent with the evidence of a functional connection between βARR1 and GLP-1 receptor, which had been previously established by Sonoda and collaborators in INS-1 β cells [30]. Another study, fully in line with our observations, has demonstrated that transgenic mice overexpressing miR-7a in pancreatic β cells developed diabetes mellitus due to impaired insulin release and β cell dedifferentiation [44]. Similarly, the specific knock-out of miR-7 in β cells increased GSIS and improved glucose homeostasis in vivo [46,47].

The key importance of miR-7 in β cell physiology is also corroborated by the fact that this miR has been shown to be one of the most abundant miRs in both human and murine islets [19], with a ratio >150 as compared with its expression between islet and surrounding acinar tissue [48].

Since βARR1 is known to play a crucial role in the desensitization of diverse GPCRs [49], our results could open the field to new research in order to verify the role of miR-7 in other tissues and cell types. βARRs can also mediate non-canonical signaling via ERK1/2 and other kinases, which are linked to β cell apoptosis [50], as well as be involved in biased agonism [51,52]. These mechanisms could be regulated by miR-7 by its direct targeting of βARR1. Intriguingly, we observed that miR-7 mimic led to a decreased GSIS as compared with scramble, even in the absence of GLP-1 stimulation (Figure 3A), suggesting that other targets of miR-7 could be involved in insulin secretion. Similarly, we cannot exclude the fact that the miR-7/βARR1 axis regulates other GPCRs in β cells. In this sense, β adrenergic receptors are known to be regulated by βARR1 [53,54,55,56]; consistent with the present findings, our group has previously demonstrated that β_2_ adrenergic receptors stimulated GSIS [57].

Our data are significant also in the clinical scenario; indeed, several drugs targeting the GLP-1 pathway are available for diabetic patients. The first group of drugs is represented by GLP-1 receptor agonists which are a group of peptides that display structural similarities to native GLP-1 and activate the GLP-1 receptor; they include exenatide, lixisenatide, albiglutide, dulaglutide, liraglutide, CJC 1134, and CJC 1131 [58]. The main adverse effects of GLP-1 receptor agonists include nausea, vomiting and diarrhea, injection-site reactions, antibody formation, and increased heart rate. The gliptins are another class of drugs indirectly acting on GLP-1, via the inhibition of GLP-1 degrading enzyme dipeptidyl peptidase 4 (DPP4). These drugs, which include sitagliptin, saxagliptin, linagliptin, alogliptin, and vildagliptin, raise the plasma levels of native GLP-1 by preventing its proteolytic degradation; some concerns have been raised regarding the metabolic and cardiovascular adverse effects of these drugs [59,60].

Notably, timing seems to be essential in the cellular responses observed by us and other investigators [3,30]. Indeed, whereas acute responses to GLP-1 (5–10 min) appear to be potentiated by knockdown of βARR1, longer-term responses (16 h) suggest a more “classical” effect of βARR1 on trafficking of the receptor with internalization suppressing downstream signaling and GSIS [3,61]. Equally important, the effects of miR-7 on the βARR1-GLP-1 axis could be different when testing lower doses of GLP-1, or when stimulating GLP-1 receptor with pharmacologic agonists.

Although miRs mainly exert their biological function to bind mRNA transcript and inhibit their translation in the cytosol [22], they have also been implicated in transcriptional gene regulation and alternative splicing, which are restricted to the cell nucleus [62,63]. Since βARR1 is known to translocate from the cytosol to the nucleus, where it regulates gene transcription [64], it would be interesting to determine the exact localization of the action of miR-7 on βARR1 within the cell.

Our study does have some limitations. First, we only conducted in vitro experiments which need to be verified in vivo in animal models. Second, we focused on βARR1, without investigating the potential contribution of other βARRs [65]. Moreover, we only used a clonal β cell line (INS-1) and we did not confirm our results in other cell lines or in human islets. Nevertheless, the GLP-1/βARR1 signaling pathway has been confirmed in murine MIN6 β cells and in human β cells [3] and we have shown that the interaction between miR-7, which is one of the most highly conserved miRs during evolution [66], and the 3′-UTR of βARR1 is conserved among species (Figure 1A). Since most of the results shown are with exogenously expressed targets or miRs, further studies are necessary to appraise the translational potential of our findings.

## Figures and Tables

**Figure 1 cells-09-01621-f001:**
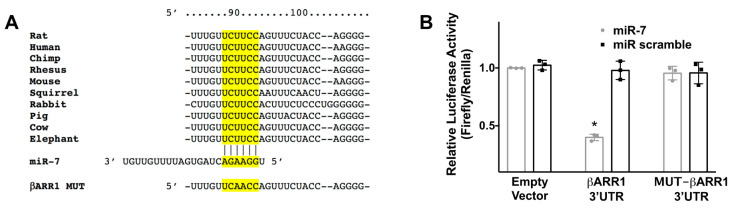
Identification of miR-7 as modulator of βARR1. (**A**) Complementary nucleotides between the target region of βARR1 3′-UTR (in yellow) and hsa-miR-7-5p (miR-7) are conserved across different species. Luciferase activity was measured 48 h after transfection, using the vector without βARR1 3′-UTR (empty vector), the vector containing the wild-type βARR1 3′-UTR, and the vector containing a mutated βARR1 3′-UTR (βARR1 MUT); (**B**) A non-targeting miR (scramble) has been employed as further control. Means ± S.E.M. are shown alongside actual values; * *p* < 0.05.

**Figure 2 cells-09-01621-f002:**
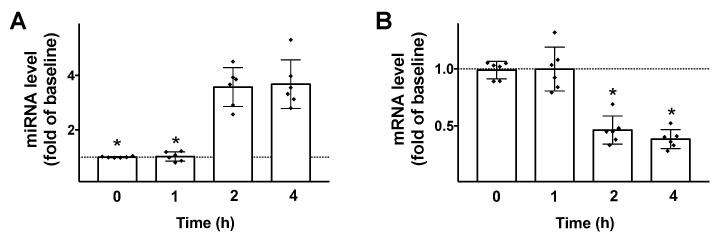
GLP-1 regulates miR-7 and βARR1 transcription in pancreatic β cells. Stimulation of INS-1 β cells with GLP-1 (100 nM) induces an upregulation of miR-7 (**A**); and a downregulation of βARR1 (**B**). Means ± S.E.M. are shown alongside actual values. * *p* < 0.05.

**Figure 3 cells-09-01621-f003:**
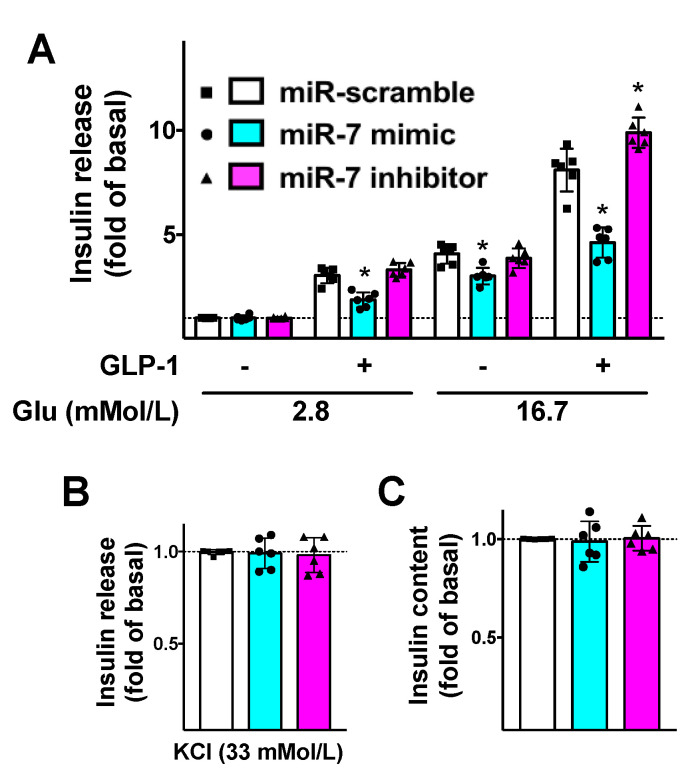
Mechanistic role of miR-7 in GLP-1-mediated insulin secretion. INS-1 cells treated with miR-7 mimic, inhibitor, or scramble (negative control) were stimulated for 2 h with GLP-1 (100 nM) or vehicle and insulin release was measured (**A**); No significant differences were observed in terms of insulin release in response to KCl (**B**) or insulin content (**C**). Means ± S.E.M. are shown alongside actual values. * *p* < 0.05 vs. miR-scramble.

**Figure 4 cells-09-01621-f004:**
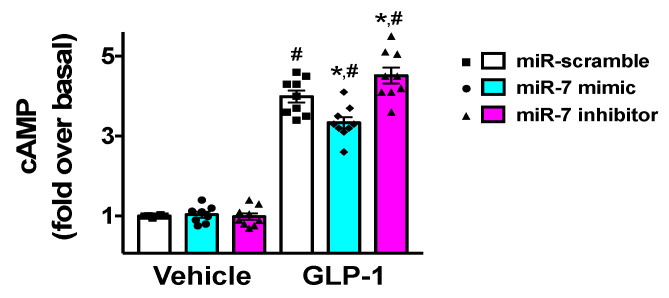
Effects of miR-7 on GLP-1-mediated cAMP production. INS-1 cells treated with miR-7 mimic, inhibitor, or scramble (negative control) were stimulated for 2 h with GLP-1 (100 nM) or vehicle and cAMP was measured. Means ± S.E.M. are shown alongside actual values. Basal, vehicle + miR-scramble; * *p* < 0.05 vs. miR-scramble, # *p* < 0.05 vs. vehicle.

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
