# Peer review of "miR-7 Regulates GLP-1-Mediated Insulin Release by Targeting β-Arrestin 1"

_cells, 2020, doi:10.3390/cells9071621_

Round 1
Reviewer 1 Report
The article by Matarrese A. et al. entitled "miR-7 Regulates GLP-1-mediated Insulin Release by 3 Targeting b Arrestin 1" is an interesting work that clarifies some new aspects of miRNAs role in GLP-1 mediated insulin secretion.
The analysis is well conducted, but probably the use of a single cell line can be a major limitation of the results and conclusions, as reported by the author himself.
In the discussion I suggest to increase the comments on the role of miRNAs in modulating pancreatic beta cells insulin secretion and I will mention all the names of GLP-1 agonists and DPP-IV inhibitors active substances commercially available.
Finally, the work is highly innovative and in my opinion worthy of publication in Cells.
Author Response
We thank this Reviewer for the words of appreciation. As requested, in the revised version of the paper, we have expanded the discussion on the role of miRs in beta cells and we have included the names of GLP-1 agonists (exenatide, lixisenatide, albiglutide, dulaglutide, liraglutide, CJC 1134 and CJC 1131) as well as DPP-IV inhibitors (sitagliptin, saxagliptin, linagliptin, alogliptin, and vildagliptin) available in the clinical field.
Reviewer 2 Report
The authors shows a function role of miR-7 in regulating GLP-1 mediated insulin release. Within the manuscript there are some comments that would be recommended for the authors to address, clarify or discuss:
- Authors should state which miR-7 they are referring to. Is it hsa-miR-7-5p?
- Understandably the authors selected miR-7 through bioinformatics screening. However, the authors should also elaborate on why only miR-7 was selected, while there should be also other miRNAs which have potential target sites to beta-arrestin-1 although yet are not selected.
- In the methods section it would be recommended for the authors to separate the topic of each method more precisely. For example in the cell culture methods section, the authors should separate the Intracellular cAMP content measurement into a section of its own.
- The authors should specify what passage these cells used were in if possible.
- Authors should provide the details of the assay used to measure miR-7 on TaqMan qPCR. Also authors should specify if the miRNA data shown was normalized.
- The authors show the sequence primers for beta-arrestin-1 and GAPDH (Supplementary Table 1) and in methods section measured on RT-qPCR, however in the results section there is no data showing the level of beta-arrestin-1 measured.
- The authors should clarify more clearly in the results presented in Figure 1B on the luciferase assay analysis, as it seems to be a bit too brief. The authors should provide the difference in the expression level of miR-7 and cel-miR-39 between the controls (wild-type), miR-7 mimic and cel-miR-39 respectively in another supplementary graph to show the effect of the miRNA mimic used in the cell samples presented for Figure 1B.
- Similarly for Figure 3, the authors should show the miR-7 levels of these samples (control, scramble samples, transfected samples with miR-7 mimic and miR-7 inhibitor in supplementary graph). The authors should show the beta-arrestin-1 levels of these GLP-1 stimulated samples as well.
- For Figure 2, the author should provide the reference for dose (100 nMol) selected GLP-1 to induce insulin release in INS1 cells. Also details of GLP-1 stimulation, the source of GLP-1 used and the stimulation procedure should be provided in the method section.
- Authors have not shown the beta-arrestin-1 levels in INS1 cells stimulated with GLP-1 at different time points as specified (in page 4 line 110).
- Authors should provide the methodology in measuring insulin release and other methods used for Figure 3 in methods section.
- It would be recommended to present all graphs as scatter plots rather than bar or floating bar graphs.
- Have authors examined the effect of overexpressing or inhibition of beta-arrestin-1 on miRNAs (such as miR-7)? It would recommended to possibly examine to understand if the effect is vice versa.
- Reference (11) refers to a long non-coding RNA and not a microRNA. The citation for this reference does not seem to be too relevant, as the sentences described refer to microRNAs in pancreatic beta cells.
- As a recommendation/suggestion to discuss, have the authors examined the localization of miR-7 and beta-arrestin-1 (whether in the nucleus or cytoplasm). Also have the authors examined if miR-7 targets beta-arrestin-1 in the nucleus or cytoplasm? Although microRNAs mainly exert function to bind mRNA transcript and inhibit their translation in the cytoplasm, it has also been implicated that microRNA are involved in transcriptional gene regulation and alternative splicing which are restricted to the cell nucleus. Beta-arrestin1 is localized in the cytoplasm and nucleus.
Author Response
The authors shows a function role of miR-7 in regulating GLP-1 mediated insulin release. Within the manuscript there are some comments that would be recommended for the authors to address, clarify or discuss:
- Authors should state which miR-7 they are referring to. Is it hsa-miR-7-5p?
Thanks for your pertinent comment. Yes, we are referring to hsa-miR-7-5p; we specify this in the revised version of the manuscript.
- Understandably the authors selected miR-7 through bioinformatics screening. However, the authors should also elaborate on why only miR-7 was selected, while there should be also other miRNAs which have potential target sites to beta-arrestin-1 although yet are not selected.
miR-7 was selected essentially for three reasons: it had a top score both in TargetScan and MirWalk3 prediction software, it is highly conserved, and it had been previously noted as a key player in beta cell physiology. Specifically, miR-7 is an evolutionarily highly conserved and is considered to be a prototypical neuroendocrine miRNA, being expressed at high levels in neurons and neuroendocrine organs, most notably the endocrine pancreas. These aspect is now included in the revised manuscript
- In the methods section it would be recommended for the authors to separate the topic of each method more precisely. For example in the cell culture methods section, the authors should separate the Intracellular cAMP content measurement into a section of its own.
We agree. We divided the methods section in 7 sub-sections, as requested.
- The authors should specify what passage these cells used were in if possible.
We now specify that cells were used between the 20th and 40th passage.
- Authors should provide the details of the assay used to measure miR-7 on TaqMan qPCR. Also authors should specify if the miRNA data shown was normalized.
We added the requested info in the methods section of the revised paper; miRNA expression was normalized to the level of U6. We apologize for the lack of details in the previous version of the manuscript.
- The authors show the sequence primers for beta-arrestin-1 and GAPDH (Supplementary Table 1) and in methods section measured on RT-qPCR, however in the results section there is no data showing the level of beta-arrestin-1 measured.
This result is shown in NEW FIGURE 2B.
- The authors should clarify more clearly in the results presented in Figure 1B on the luciferase assay analysis, as it seems to be a bit too brief. The authors should provide the difference in the expression level of miR-7 and cel-miR-39 between the controls (wild-type), miR-7 mimic and cel-miR-39 respectively in another supplementary graph to show the effect of the miRNA mimic used in the cell samples presented for Figure 1B. Similarly for Figure 3, the authors should show the miR-7 levels of these samples (control, scramble samples, transfected samples with miR-7 mimic and miR-7 inhibitor in supplementary graph). The authors should show the beta-arrestin-1 levels of these GLP-1 stimulated samples as well.
We agree with the Reviewer and we apologize for the confusion and the lack of details in the methods section of the previous version of the manuscript. We have repeated the experiment using a commercially available pmirGLO Dual-Luciferase miRNA Target Expression Vector (Promega, Madison, WI, USA). We transfected the cells with 0.05 μg of the plasmid (vector alone, bARR and mutated bARR), and with 50 nM of miR-7 or non-targeting miR (negative control), using the same quantities in all groups. The results are shown in the NEW FIGURE 1B.
Moreover, we are now showing in the NEW SUPPLEMENTARY FIGURE 1 the suppression of bARR1 levels in b cells treated with miR-7 mimic, as requested by this Reviewer.
We hope that the revised methods and the new figure are now clear and satisfactory.
- For Figure 2, the author should provide the reference for dose (100 nMol) selected GLP-1 to induce insulin release in INS1 cells. Also details of GLP-1 stimulation, the source of GLP-1 used and the stimulation procedure should be provided in the method section.
We specify that the effective dose of GLP-1 had been shown by Sonoda et al. 10.1073 pnas.0710402105.
GLP-1 was obtained from Millipore-Sigma; in the methods section, we specify that all reagents were from Millipore-Sigma (Burlington, MA, USA), unless otherwise stated.
- Authors have not shown the beta-arrestin-1 levels in INS1 cells stimulated with GLP-1 at different time points as specified (in page 4 line 110).
This result is shown in FIGURE 2B.
- Authors should provide the methodology in measuring insulin release and other methods used for Figure 3 in methods section.
Thanks. This methodology has been added.
- It would be recommended to present all graphs as scatter plots rather than bar or floating bar graphs.
Done.
- Have authors examined the effect of overexpressing or inhibition of beta-arrestin-1 on miRNAs (such as miR-7)? It would recommended to possibly examine to understand if the effect is vice versa.
This is intriguing. However, even if we show that overexpressing or inhibition of beta-arrestin-1 could be associated with modifications of miRNA, our take-home message (miR-7 targets beta-arrestin 1) does not change and, more importantly, we will not have an exact mechanism to prove how beta-arrestin 1 could regulate the levels of miRNAs in beta cells (regulation of gene transcription, of protein involved in miRNA generation, of miRNA degradation?). Again, this aspect is interesting, but we respectfully believe that it deserves to be investigated in a dedicated project.
- Reference (11) refers to a long non-coding RNA and not a microRNA. The citation for this reference does not seem to be too relevant, as the sentences described refer to microRNAs in pancreatic beta cells.
Ref.11 has been removed, as requested.
- As a recommendation/suggestion to discuss, have the authors examined the localization of miR-7 and beta-arrestin-1 (whether in the nucleus or cytoplasm). Also have the authors examined if miR-7 targets beta-arrestin-1 in the nucleus or cytoplasm? Although microRNAs mainly exert function to bind mRNA transcript and inhibit their translation in the cytoplasm, it has also been implicated that microRNA are involved in transcriptional gene regulation and alternative splicing which are restricted to the cell nucleus. Beta-arrestin1 is localized in the cytoplasm and nucleus.
We thank this Reviewer for her/his comment. We discuss this intriguing aspect in the revised version of the paper.
Reviewer 3 Report
Matarese and colleagues extend previous work on the role of bARR1 in regulating GLP1R function in beta-like cells in vitro. They use a bioinformatic approach to discover naturally occurring miRs that could suppress bARR1 expression and discover miR7 to have this function. They show that a mimic of this miR can indeed suppress bARR1 expression in a presumably GLP-1R expressing pancreatic beta cell-line, and subsequently that it suppresses glucose stimulated insulin secretion (GSIS) by these cells. This data is supported by previous findings that knockdown of bARR1 using siRNA also suppressed GSIS in a similar cell-line (Sonoda et al. 10.1073 pnas.0710402105). The mechanism for this, however, remains rather complex since timing seems to be essential. Acute responses to GLP-1 (5-10 minutes) seem to be potentiated by knockdown of bARR1, but longer-term responses (16 hours) seems to suggest a more “classical” effect of bARR1 on trafficking of the receptor with internalization suppressing down-stream signaling and GSIS (Jones et al. 10.1038/s41467-018-03941-2).
The paper is relatively clearly written and presented. However, I still have some comments.
- The Materials and Methods I find very minimal and seem to leave out some important information regarding the set-up of the experiments:
- Mention is made of the effect of GLP1 on bARR1 gene expression (and primer sequences are provided), but no data is shown or described. It would be useful to know if GLP1 caused a concomitant increase in miR7 and suppression of bARR1. At the same time, was there any effect on bARR2 gene expression?
- In the transfection experiments – are cells transiently or stably transfected with the miR expression constructs? If transient, how long after transfection were the cells used in agonist-induced experiments.
- Why was 2 hours used for assessing cAMP and GSIS responses to GLP-1?
- miR7 mimic overexpression causes what seems to be a relatively minor suppression of the cAMP response to GLP-1 (fig. 4, -10%), but are more substantial effect on GSIS (-40%). The authors should comment on this. In relation to this, did miR7 overexpression also suppress the pERK1/2 and pCREB responses to GLP-1 described in Sonoda et al.?
Author Response
Matarese and colleagues extend previous work on the role of bARR1 in regulating GLP1R function in beta-like cells in vitro. They use a bioinformatic approach to discover naturally occurring miRs that could suppress bARR1 expression and discover miR7 to have this function. They show that a mimic of this miR can indeed suppress bARR1 expression in a presumably GLP-1R expressing pancreatic beta cell-line, and subsequently that it suppresses glucose stimulated insulin secretion (GSIS) by these cells. This data is supported by previous findings that knockdown of bARR1 using siRNA also suppressed GSIS in a similar cell-line (Sonoda et al. 10.1073 pnas.0710402105). The mechanism for this, however, remains rather complex since timing seems to be essential. Acute responses to GLP-1 (5-10 minutes) seem to be potentiated by knockdown of bARR1, but longer-term responses (16 hours) seems to suggest a more “classical” effect of bARR1 on trafficking of the receptor with internalization suppressing down-stream signaling and GSIS (Jones et al. 10.1038/s41467-018-03941-2).
The paper is relatively clearly written and presented.
We thank this Reviewer for her/his words of appreciation.
However, I still have some comments.
The Materials and Methods I find very minimal and seem to leave out some important information regarding the set-up of the experiments:
Mention is made of the effect of GLP1 on bARR1 gene expression (and primer sequences are provided), but no data is shown or described. It would be useful to know if GLP1 caused a concomitant increase in miR7 and suppression of bARR1. At the same time, was there any effect on bARR2 gene expression?
We thank this Reviewer for the pertinent comments. We have expanded the methods section.
We now provide a new figure showing the effect of GLP1 on bARR1, as requested.
According to both Targetscan and miRwalk in silico prediction tools, the seed region of miR-7 is not complementary to the 3’UTR of bARR2; therefore, we do not expect that bARR2 gene expression is regulated by miR-7.
In the transfection experiments – are cells transiently or stably transfected with the miR expression constructs? If transient, how long after transfection were the cells used in agonist-induced experiments.
We thank this Reviewer for this comment and we apologize for the lack of details in the previous version of the manuscript. We now specify that the cells were transiently transfected and that the experiments were performed forty-eight hours after transfection.
Why was 2 hours used for assessing cAMP and GSIS responses to GLP-1?
We used the 2-hour time point in order to reproduce the experimental conditions published by Sonoda et al. (Fig. 4 and 5; 10.1073 pnas.0710402105). Nevertheless, we added a brief paragraph in the discussion to address the importance of timing when assessing the relationship between beta arrestin and GLP-1.
miR7 mimic overexpression causes what seems to be a relatively minor suppression of the cAMP response to GLP-1 (fig. 4, -10%), but are more substantial effect on GSIS (-40%). The authors should comment on this. In relation to this, did miR7 overexpression also suppress the pERK1/2 and pCREB responses to GLP-1 described in Sonoda et al.?
We thank this Reviewer for this comment. As requested, we now provide (NEW SUPPLEMENTARY FIGURE 2) immunoblots assessing pERK1/2 and pCREB responses to GLP-1, overall confirming the findings reported by Sonoda et al.
Reviewer 4 Report
This is an interesting study with the results clearly shown. There is however no validation of the results with endogenous targets, as admitted by the authors. Therefore one has to remain cautious about the extent of the interpretation that can be made of these results. Some of the claims made by the authors need to be therefore toned down as most of the results shown are with exogenously expressed targets and/or miRs.
Author Response
We thank this Reviewer for her/his comment. We agree and we added a sentence in the discussion in order to address this limitation.
Round 2
Reviewer 2 Report
The authors have addressed the comments and much improved the manuscript with elaboration on the methods and discussion, along with additional supplementary data. There are a few minor comments recommended:
In Figure 3A, a significant decrease in insulin release compared to the negative control (scramble) is observed in the vehicle (without GLP-1 induced), therefore with only a higher dose of (16.7 mMol/L) glucose in miR-7 mimic. I would be recommended for the authors to mention and explain the potential reason for this effect observed.
Also in Supplementary Figure 1, it is interesting that the inhibition of miR-7 does not show significant effect compared to control (miR scramble) on upregulating beta-arrestin 1 expression (after 48 hours?). Could the authors please address why this potentially may be the case. Also it would be recommended to mention/iterate the exposure time of miR-7 mimic, inhibitor or scramble in legend.
The authors should provide the TaqMan microRNA assay IDs for the miR-7 and U6 assays used.
Some of the sentences such as in lines 215 to 219 seem too long and would be recommended to split them into two sentences.
Would be recommended also to be consistent with the using of miR or miRNA. Likewise the term INS-1 beta cells, INS-1 cells or pancreatic beta cells; and also GLP-1 or GLP1.
In line 88 and 89, italics is used for “INS-1” cells and “two”, this should not be in italics. Also WT (abbreviation) should be described as wild type (WT).
Author Response
We addressed the minor issues raised by Reviewer #2.